# Tukey $g$-and-$h$ neural network regression for non-Gaussian data

## Abstract

This paper addresses non-Gaussian regression with neural networks via the use of the Tukey $g$-and-$h$ distribution. The Tukey $g$-and-$h$ transform is a flexible parametric transform with two parameters $g$ and $h$ which, when applied to a standard normal random variable, introduces both skewness and kurtosis, resulting in a distribution commonly called the Tukey $g$-and-$h$ distribution. Specific values of $g$ and $h$ produce good approximations to other families of distributions, such as the Cauchy and student's $t$ distributions. The flexibility of the Tukey $g$-and-$h$ distribution has driven its popularity in the statistical community, in geosciences and finance. In this work we consider the training of a neural network to predict the parameters of a Tukey $g$-and-$h$ distribution in a regression framework via the minimization of the corresponding negative log-likelihood, despite the latter having no closed-form expression. We demonstrate the efficiency of our procedure in simulated settings and present an application to a real-world dataset of global crop yield for several types of crops. Finally, we assess our probabilistic predictions via the logarithmic score and Probability Integral Transform. A PyTorch implementation is made available on GitHub and as a PyPI package.

## 1 Introduction

The application of Deep Learning to highly non-linear regression tasks has brought unprecedented results in many real-world applications. In many practical regression problems, it is desirable to predict a probabilistic distribution rather than a single-point value. This is key when machine learning algorithms are used in complex decision-making processes. For instance, if a neural network is used to predict a stock price or the duration of a surgical operation (Ng et al., 2017), confidence intervals will be far more valuable than single point predictions. One must distinguish between different sources of uncertainty: model uncertainty — which we do not consider as it is not the main topic of this paper but which could be addressed via Bayesian neural networks (Neal, 2012) or Bootstrapping (Lee et al., 2020) — and what is often referred to as aleatoric uncertainty, that is, the remaining randomness even if the true regression model was known exactly. We note that inherent uncertainty may not only arise from the generating process, but also from the potentially limited set of features used for regression. One common approach to account for aleatoric uncertainty consists in predicting, for a given set of features, the two parameters of a Gaussian distribution, rather than a single-point value. This is particularly valuable in the case where the conditional variance of the target also depends on the features. However, the Gaussian assumption might be quite restrictive, as it does not allow for skewness and kurtosis of the target variable conditioned on the features. These patterns may arise in the target variable either from the generating process itself, or from using an incomplete set of features. One way to address non-Gaussianity in the target variable is to consider more flexible parametric families of distributions. For instance, one might train a neural network to output the parameters of a mixture of Gaussian distributions rather than a single Gaussian distribution. This might be useful in particular when the conditional distribution of the target is multi-modal, but remains limited, in particular when it comes to modelling heavy tails. It is also worth mentioning non-parametric approaches such as quantile regression (Xu et al., 2017; Taylor, 2000). In this paper, however, we focus on the parametric approach, more specifically on the Tukey $g$-and-$h$ probability distribution (Jorge & Boris, 1984). This distribution is obtained by applying

the Tukey $g$-and-$h$ transform with parameters $g$ and $h$ to a standard normal random variable, which is then rescaled by a multiplicative factor $\sigma$ and shifted by an additive constant $\mu$. As such, the Tukey $g$-and-$h$ distribution has four parameters. The Tukey $g$-and-$h$ distribution has been widely used in modern statistical geosciences (Xu & Genton, 2017; Jeong et al., 2019), spatio-temporal modelling (Murakami et al., 2021), but also in financial risk analysis (M. Bee & Trapin, 2021). The standard random field approach (Xu & Genton, 2017) consists in modelling spatio-temporal dependence via a Gaussian Process, followed by a pointwise mapping by the Tukey $g$-and-$h$ transform to incorporate non-Gaussian patterns. In parallel, the application of neural networks to geosciences has grown exponentially, for instance to address problems such as hyper-resolution (Höhlein et al., 2020), forecasting (Ren et al., 2021) or for the parameterization of discretized non-linear PDE solvers (Guillaumin & Zanna, 2021) and yield forecasting (You et al., 2017), (Dado et al., 2020), (Wang et al., 2020). The objective of this work is to propose a method that bridges the gap between Tukey $g$-and-$h$ random fields and neural network regression. In particular, we see a longer-term incentive to combine neural networks to learn complex feature-dependent parameters of a Tukey $g$-and-$h$ transform with Gaussian Process models to incorporate residual patterns of spatio-temporal dependence.

Our paper is organized as follows. In Section 2, we review the Tukey $g$-and-$h$ transform and present our methodology for training and evaluating neural networks for the prediction of $g$-and-$h$ distributions. This requires us to obtain the derivatives of the Tukey $g$-and-$h$ log likelihood with respect to the parameters, which involves the inversion of the Tukey $g$-and-$h$ transform. There is no known closed form for the latter. We propose the use of binary search to address this issue, which is efficient both numerically and computationally. In Section 3, we demonstrate the benefits of the proposed methodology based on simulated data experiments, while in Section 4 we present an application to a real-world regression problem by applying our methodology to learn patterns of global yield for several crops. Finally, we provide a PyTorch implementation of the Tukey $g$-and-$h$ loss function made publicly available on the GitHub account of the authors.

## 2 Methodology

In this section, we present our proposed methodology for the use of the Tukey $g$-and-$h$ transform in neural network regression. We first review the Tukey $g$-and-$h$ transform and its basic properties. We then consider the evaluation of the negative log-likelihood function, which we will use as our loss function for training. In evaluating the negative log-likelihood, the main challenge lies in the inversion of the Tukey $g$-and-$h$ transform, which is required to evaluate the probability density function of the Tukey $g$-and-$h$ distribution. We propose to use binary search to address this problem. In comparison to other approaches proposed to approximate the inverse of the Tukey $g$-and-$h$ transform such as grid search, this entails no approximation other than that incurred by numerical precision. Finally, we discuss how one can assess goodness-of-fit and obtain confidence intervals in Tukey $g$-and-$h$ neural network regression.

### 2.1 The Tukey $g$-and-$h$ transform

We review some properties of the Tukey $g$-and-$h$ transform (Xu & Genton, 2017; Jorge & Boris, 1984) and its ability to approximate a wide range of other well-known families of probability distributions when applied to a standard normal random variable, such as student's $t$-distribution.

The Tukey $g$-and-$h$ transform with parameters $g \in \mathbb{R}$ and $h \geq 0$ is defined by,

$$\tau_{g,h}(z) = \frac{\exp(gz) - 1}{g} \exp\left(\frac{1}{2}hz^2\right), \quad \forall z \in \mathbb{R}, \tag{1}$$

when $g \neq 0$. The case $g = 0$ is obtained by continuous extension,

$$\tau_{0,h}(z) = z \exp\left(\frac{1}{2}hz^2\right), \quad \forall z \in \mathbb{R}.$$

When both $g$ and $h$ are zero, the transform is just the identity function. For fixed $g$ and $h$ the transform $\tau_{g,h}(z)$ is an increasing function in $z$.

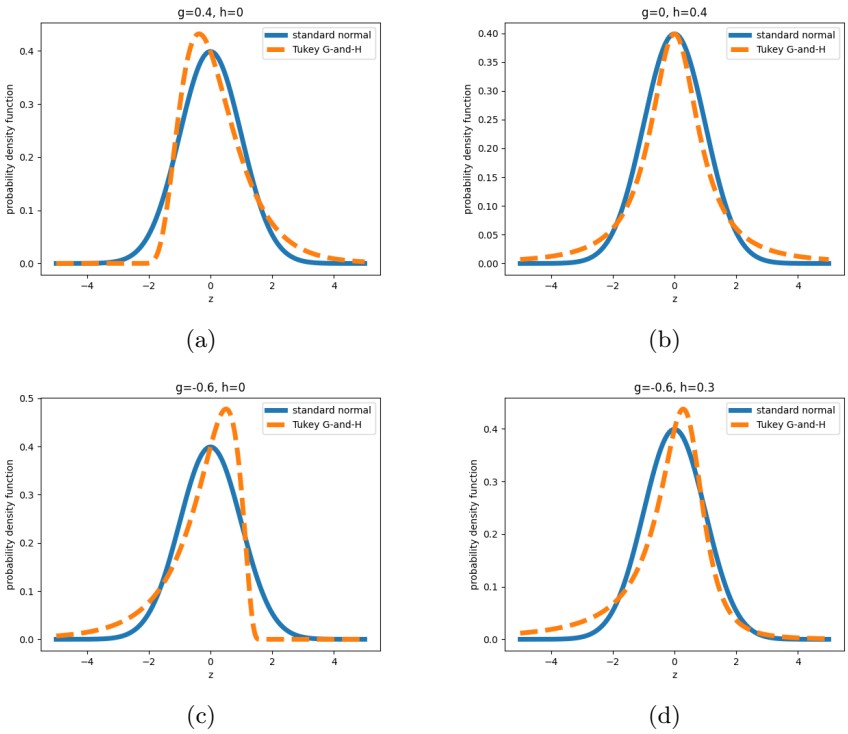

Figure 1: Comparison of the probability density function of a Tukey $g$-and-$h$ distributed random variable with that of a standard normal random variable, for different values of $g$ and $h$.

Let $Z$ be a standard normal random variable, and define,

$$\begin{cases} \widetilde{Z} & = \tau_{g,h}(Z), \\ Y & = \mu + \sigma\widetilde{Z}. \end{cases} \tag{2}$$

where $\mu \in \mathbb{R}$ and $\sigma \in \mathbb{R}$ roughly control the first two moments of the transformed random variable $Y$. The parameters $g$ and $h$ roughly control, respectively, the skewness and kurtosis, which are zero for the non-transformed variable $Z$. A positive value of $g$ will incur positive skewness in $Y$ — see Figure 1a— while a negative value of $g$ will incur negative skewness — see Figure 1c. Note that as $g$ and $h$ converge to zero, $Y$ converges in distribution towards a Gaussian random variable with mean $\mu$ and standard deviation $\sigma$. In this manuscript, we will say that the transformed random variable $\widetilde{Y}$ follows a g-and-g distribution. In Figure 1 we show the probability density function of $\widetilde{Z}$ for 4 combinations of the parameters $g$ and $h$, superimposed with the probability density function of a standard normal distribution. The Tukey $g$-and-$h$ distribution provides a good approximation to a wide range of standard probability distributions (Jorge & Boris, 1984).

## 2.2 Training via negative likelihood loss minimization

We consider a regression problem with features $\mathbf{X}$ and target $Y$. Rather than only training a neural network to predict the conditional expectation of $Y$ given $\mathbf{X}$, we wish to approximate the full conditional distribution of $Y$ given $\mathbf{X}$ by training the neural network to predict a $g$-and-$h$ distribution — that is, predict in terms of $\mathbf{X}$ the four parameters $\mu, \sigma, g$ and $h$ of such a distribution. As such, the output of our neural network should have a total of 4 neurons, one for each parameter of a $g$-and-$h$ distribution. In this section, we provide details on how to train a neural network for Tukey $g$-and-$h$ regression via stochastic gradient descent on the negative log-likelihood. Following standard properties of transformed random variables, the transformed random variable $Y$ defined in (2) admits a probability density function which can be expressed in terms of the probability density function $f_Z(z) = \frac{1}{\sqrt{2\pi}}\exp(-z^2/2)$ of the standard normal random variable $Z$.

Specifically,

$$f_Y(y) = \frac{1}{\sigma \; \tau'_{g,h}\left(\tau^{-1}_{g,h}\left(\frac{y-\mu}{\sigma}\right)\right)} f_Z\left(\tau^{-1}_{g,h}\left(\frac{y-\mu}{\sigma}\right)\right),$$

where $\tau'_{g,h}(z)$ is the derivative of the Tukey $g$-and-$h$ transform $\tau_{g,h}(\cdot)$ given by,

$$\tau'_{g,h}(z) = \left[\exp(gz) + hz\frac{\exp(gz)-1}{g}\right]\exp\left(\frac{1}{2}hz^2\right),$$

and $\tau^{-1}_{g,h}$ is the inverse function of the Tukey $g$-and-$h$ transform for fixed parameters $g$ and $h$.

Given a collection of features $\mathbf{x}_i$ and response variables $y_i$, $i = 1,\ldots,n$, we denote $\mu_i, \sigma_i, g_i$ and $h_i$ the 4 components of the neural network's output for input $\mathbf{x}_i$, corresponding to the four parameters of a $g$-and-$h$ distribution. If we introduce

$$\widehat{z}_i = \tau^{-1}_{g_i,h_i}\left(\frac{y-\mu_i}{\sigma_i}\right), \tag{3}$$

the negative log-likelihood function can be expressed in terms of the neural network's parameters $\theta$ as,

$$l(\theta) = \sum_i \log \tau'_{g_i,h_i}(\hat{z}_i) + \sum_i \log \sigma_i + \frac{1}{2}\sum_i \hat{z}_i^2$$

$$= \sum_i \log\left[\exp(g\hat{z}_i) + h\hat{z}_i\frac{\exp(g\hat{z}_i)-1}{g}\right] + \sum_i \log \sigma_i + \sum_i \frac{1+h_i}{2}\hat{z}_i^2. \tag{4}$$

The dependence of $g_i$, $h_i$, $\sigma_i$ and $\hat{z}_i$ on the neural network's parameters $\theta$ is left implicit in the notation for simplicity. The evaluation of the $\hat{z}_i$'s and their gradient with respect to the parameters $g$ and $h$ — these are required to carry out Stochastic Gradient Descent — poses the main challenge to the evaluation of the loss. One previously proposed approach is to approximate the inverse Tukey $g$-and-$h$ transform using a grid of values $z_0,\ldots,z_p$ for which $\tau_{g,h}$ is evaluated (Xu & Genton, 2017). In contrast, we propose an approach which is not limited in accuracy other than by that of the precision of the numerical implementation. More specifically, since the Tukey $g$-and-$h$ transform is an increasing function of $z$, its inverse at any given point can be obtained efficiently by binary search. The only requirement is that we start the algorithm with a large enough range. We note that this binary search need only be applied once for every feedforward call of the stochastic gradient algorithm.

Finally, in order to apply stochastic gradient descent algorithms, we also require the derivatives of the inverse transform with respect to $g$, $h$ and $\widetilde{z}$ due to the terms $\hat{z}_i$ in the loss function (4). First, we have,

$$\frac{\partial \tau^{-1}_{g,h}}{\partial \widetilde{z}}(\widetilde{z}) = \frac{1}{\tau'_{g,h}(\tau^{-1}_{g,h}(\widetilde{z}))}. \tag{5}$$

Then, we derive,

$$\frac{\partial \tau_{g,h}}{\partial g}(z) = \frac{\exp(gz)(gz-1)+1}{g^2}\exp(\frac{1}{2}hz^2), \tag{6}$$

$$\frac{\partial \tau_{g,h}}{\partial h}(z) = \frac{z^2}{2}\tau_{g,h}(z). \tag{7}$$

We then write,

$$\tau_{g,h}\left(\tau^{-1}_{g,h}(\widetilde{z})\right) = \widetilde{z}, \tag{8}$$

and obtain the desired quantities after taking the derivative of the above equation with respect to $g$ and $h$ respectively, by application of the chain rule,

$$\frac{\partial \tau^{-1}_{g,h}}{\partial g}(\widetilde{z}) = -\frac{\frac{\partial \tau_{g,h}}{\partial g}\left(\tau^{-1}_{g,h}(\widetilde{z})\right)}{\tau'_{g,h}(\tau^{-1}_{g,h}(\widetilde{z}))}, \tag{9}$$

$$\frac{\partial \tau^{-1}_{g,h}}{\partial h}(\widetilde{z}) = -\frac{\frac{\partial \tau_{g,h}}{\partial h}\left(\tau^{-1}_{g,h}(\widetilde{z})\right)}{\tau'_{g,h}(\tau^{-1}_{g,h}(\widetilde{z}))}. \tag{10}$$

By defining a *Function* object in PyTorch that encapsulates the inverse Tukey transform and its derivatives with respect to $g$, $h$ and $\widetilde{z}$, we can obtain the gradient of the negative log-likelihood function via automatic differentiation. The code provided alongside this manuscript takes care of implementing these computations, so that the user only needs to declare a TukeyGandHLoss object and provide the four parameters output by their neural network for each data point of a mini-batch.

In Table 6 in the Appendix, we provide computational times averaged over 1000 values of running one step of the Adam algorithm (Kingma & Ba, 2015) for the Gaussian and Tukey g-and-h losses. We do so for a fully connected forward neural network for varying number of layers and layer size. For small neural networks, the computational cost of the Tukey g-and-h loss is approximately 10 times that of the Gaussian loss. However as the depth and width of the neural network increase, the additional cost incurred by the Tukey g-and-h loss becomes negligible compared to the cost of the backward propagation, resulting in similar overall computational times between the two methods.

## 2.3 Prediction intervals

One benefit of the Tukey *g*-and-*h* distribution for the modelling of non-Gaussian random variables is that we can easily obtain prediction intervals for the target variable. Let $0 < \alpha < 1$. Due to the continuous and increasing nature of the Tukey *g*-and-*h* transform, we immediately obtain,

$$F_{\widetilde{Z}}^{-1}(\alpha) = \tau_{g,h}(\Phi^{-1}(\alpha)), \tag{11}$$

where $F_{\widetilde{Z}}^{-1}$ is the inverse cumulative distribution function of the transformed random variable, and $\Phi^{-1}$ is the inverse cumulative distribution function of the standard normal distribution.

Denote $\widehat{\theta}$ the parameters of the trained neural network. For an input $\mathbf{x}$, write

$$\mu(\mathbf{x}; \widehat{\theta}), \sigma(\mathbf{x}; \widehat{\theta}), g(\mathbf{x}; \widehat{\theta}) \text{ and } h(\mathbf{x}; \widehat{\theta}) \tag{12}$$

for the four parameters of the *g*-and-*h* distribution output by the neural network with parameters $\widehat{\theta}$ when provided with input $\mathbf{x}$. An $\alpha$-level confidence interval for a feature $\mathbf{x}$ is provided by,

$$\left[ \mu(\mathbf{x}; \widehat{\theta}) + \sigma(\mathbf{x}; \widehat{\theta}) \tau_{g(\mathbf{x}; \widehat{\theta}), h(\mathbf{x}; \widehat{\theta})}(-z_{1-\alpha/2}) \, , \right.$$
$$\left. \mu(\mathbf{x}; \widehat{\theta}) + \sigma(\mathbf{x}; \widehat{\theta}) \tau_{g(\mathbf{x}; \widehat{\theta}), h(\mathbf{x}; \widehat{\theta})}(z_{1-\alpha/2}) \right],$$

with $z_\alpha \equiv F_Z^{-1}(\alpha)$. While this approach is simple and computationally efficient, it was noted in (Xu & Genton, 2017) that it may lead to unreasonably large prediction intervals when the skewness parameter $g$ is large in absolute value. Following their approach, one might instead consider the following prediction interval,

$$\left[ \mu(\mathbf{x}; \widehat{\theta}) + \sigma(\mathbf{x}; \widehat{\theta}) \tau_{g(\mathbf{x}; \widehat{\theta}), h(\mathbf{x}; \widehat{\theta})}(-z_{1-\alpha+\gamma}) \, , \right.$$
$$\left. \mu(\mathbf{x}; \widehat{\theta}) + \sigma(\mathbf{x}; \widehat{\theta}) \tau_{g(\mathbf{x}; \widehat{\theta}), h(\mathbf{x}; \widehat{\theta})}(z_{1-\gamma}) \right],$$

where $0 \le \gamma \le \alpha$ is chosen so that it minimizes the length of the prediction interval. We note that the prediction intervals obtained in this fashion do not account for the uncertainty in $\widehat{\theta}$. While this is not the focus of this paper, a potential means of addressing this issue might be the use of bootstrapping (Sluijterman et al., 2023).

## 2.4 Quantitative and qualitative metrics for probabilistic prediction

The general framework of this work is that of probabilistic prediction: we wish to approximate the conditional distribution $p(y|x)$ of an observation $y$ given features $x$ via a parametric distribution whose parameters are controlled by the output of a neural network. In order to assess the skill of the trained neural network on

this task, we consider several qualitative and quantitative metrics, based on the litterature on probabilistic prediction.

As a quantitative metric, we consider the negative logarithm of the probability density function evaluated at the observations. In the context of probabilistic prediction, this is referred to as Logarithmic Score (LS) Gneiting & Katzfuss (2014). If we write $p(y; \widehat{\theta}(x))$ for the probability density function of the predicted distribution at features $x$, with $\widehat{\theta}$ the parameters of the distribution output by the neural network, the Logarithmic Score over a dataset $(x_i, y_i)_{i=1,\dots,n}$ is,

$$LS = -\frac{1}{n} \sum_{i=1}^{n} \ln p(y_i; \widehat{\theta}(x_i)). \tag{13}$$

To compare two trained distributions, say $p_A(y; \widehat{\theta_A}(x))$ and $p_B(y; \widehat{\theta_B}(x))$, the mean log likelihood ratio is nothing else than the difference in the logarithmic scores,

$$\Lambda(A, B) = \frac{1}{n} \sum_{i=1}^{n} \ln \frac{p_A(y; \widehat{\theta_A}(x))}{p_B(y; \widehat{\theta_B}(x))} = LS_B - LS_A. \tag{14}$$

The sign of the above quantity will indicate which model performs better. The absolute value of this quantity also has a natural interpretation: roughly speaking, if $\Lambda(A, B) > 0$, for instance, the probability of a single observation under $\widehat{p}_A$ is, on average, $\exp(\Lambda(A, B))$ higher than under $\widehat{p}_B$.

While the Logarithmic Score is a useful quantity to compare two models, for a given model it cannot indicate a potential misfit with the data. For this purpose, following Gneiting & Katzfuss (2014), we leverage the Probabilistic Integral Transform (PIT). Given an observation $(x, y)$ and a predicted cumulative distribution function $\widehat{F}_Y$, the PIT is defined as,

$$PIT(\widehat{F}_Y, y) = F_Y(y; \widehat{\theta}(x)). \tag{15}$$

If the observations are distributed according to the predicted distribution, the PIT values should follow a uniform distribution on the interval $[0, 1]$. Finally, we may also assess the quantiles of the predictive distribution by defining,

$$q[100\alpha] = \frac{1}{n} \sum_{i=1}^{n} \mathbb{1}(y_i \leq F_Y^{-1}(\alpha; \widehat{\theta}(x_i))), \quad 0 < \alpha < 1, \tag{16}$$

where $F_Y^{-1}$ is the inverse cumulative distribution.

## 3 Simulated data experiments

In this section we present some results on Tukey $g$-and-$h$ neural network regression in simulated settings. We first study the case where the true distribution of the target variable, conditioned on the features, is indeed a Tukey $g$-and-$h$ distribution. We then assess the robustness of the proposed methodology in a misspecified setting where the target variable follows a student's $t$-distribution, conditionally on the features.

### 3.1 No model misspecification

We first consider the case where the true conditional distribution of the target variable $Y$ actually belongs to the family of Tukey $g$-and-$h$ distributions. Specifically, we consider a scalar feature $X$ uniformly distributed on the interval $[0, 1]$,

$$X \sim \mathcal{U}([0, 1]). \tag{17}$$

The response variable $Y$ conditionally on $X$ is written as,

$$Y|X \sim \sigma(X)\tau_{g(X),h(X)}(Z) + \mu(X), \tag{18}$$

with $Z \sim \mathcal{N}(0, 1)$. For the purposes of this simulated data experiment, the deterministic functions $\sigma(\cdot)$, $\mu(\cdot)$, $g(\cdot)$ and $h(\cdot)$ are set to arbitrarily chosen closed-form functions, available in the online code. From these definitions, we simulate 10 independent datasets of size 4000, and 10 of size 40,000.

We define a fully-connected neural network with a single input, corresponding to the scalar feature $X$. The number of outputs depends on the probabilistic distribution: 4 for the Tukey $g$-and-$h$ distribution, and 2 for the Gaussian distribution. We optimize hyperparameters (see Appendix A) — number of layers, size of layers, initial learning rate, step size of the learning rate scheduler, batch size and use of batch normalization — using the Optuna Akiba et al. (2019) framework over 50 trials for each dataset, for both the Gaussian and Tukey $g$-and-$h$ distributions. For each of the 50 trials, we use 80% of the data for training and 10% for validation. The remaining 10% of the data are used as test data for the best trial from the hyperparameter optimization procedure.

In Figure 2 we show the resulting fit between the true regression functions and the functions learnt by the neural network for one particular dataset. In Figure 3 we compare the fitted Tukey $g$-and-$h$ distribution, and a fitted Gaussian distribution for different values of the feature $X$. Unsurprisingly, we observe that in the case of a left skew of the target variable (e.g. x=0.1, top left), the mode of the fitted Gaussian distribution is underestimated. On the contrary, in the case of a right skew of the target variable (e.g. x=0.9, bottom right), the mode of the fitted Gaussian distribution is overestimated. We additionally report summary statistics of the validation and test Logarithmic Scores aggregated over the 10 independent datasets, please see Appendix B. These confirm what one would expect: the fit provided by the Tukey g-and-h distribution is better than that provided by a Gaussian distribution.

## 3.2 Model misspecification

We now present a simulated data experiment subject to model misspecification. Specifically, we set the target variable to follow a $t$-distribution, the parameters of which are controlled by arbitrarily specified functions,

$$Y|X \sim \mu(X) + \sigma(X)T, \quad T \sim t_{\nu(X)}, \tag{19}$$

where $t_\nu$ is used to denote student's $t$-distribution with $\nu$ degrees of freedom. Naturally, if we know the true distribution of the target variable to follow a $t$-distribution, we are likely to be better off directly using that in our regression model. Here our intent is to show that our procedure is robust to a misspecified setting. As we mentionned earlier, the Tukey $g$-and-$h$ provides a good approximation to student's $t$-distribution. It is therefore natural to expect from our method that it should be robust to this slight departure from the model distribution. In stronger cases of misspecification, such as that of a multi-modal distribution, there is no reason to expect good performance from our Tukey $g$-and-$h$ approach in general. One could however adapt Mixture Density Networks (Bishop, 1994) to mixtures of Tukey $g$-and-$h$ distributions. While in principle, as mentioned by their author, Mixture Density Networks can approximate any distribution, the number of required components in the mixture may be very large for heavy tailed distributions, an issue that could be addressed by replacing the Gaussian components of the mixture with Tukey $g$-and-$h$ components. In Figure 4 we compare the fitted distribution for four values of the scalar input. The Tukey $g$-and-$h$ recovers the shape of the distribution of the target variable, in comparison to the Gaussian neural network regression.

## 4 Real data experiment: an application to crop yield predictions

Food security is widely recognised as one of the most urgent challenges we currently face globally (Intergovernmental Panel on Climate Change (IPCC), 2023). This concern has grown in significance due to the changing climate and its warming effects. With the increasing occurrence of extreme weather events and alterations in weather patterns, our food system, especially in certain regions, has become highly vulnerable to the impacts of climate change. Agriculture is both one of the sectors most susceptible to climate change and a significant contributor to it (Mahowald et al., 2017). Therefore, it is essential to consider both mitigation and adaptation strategies, as well as transforming agricultural practices to promote sustainability and resilience. A key objective is to develop more reliable and scalable methods for monitoring global crop conditions promptly and transparently, while also exploring how we can adapt agriculture to mitigate the effects of climate change.

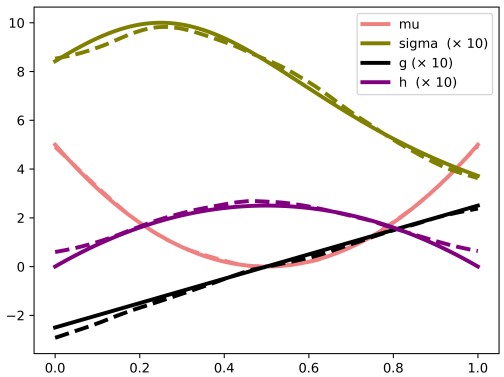

Figure 2: True parameter functions (solid) and trained parameter functions (dashed) on 40000 observations.

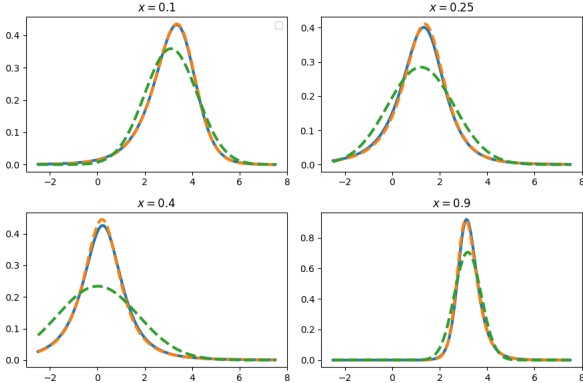

Figure 3: Comparison of true (blue) and fitted (Tukey $g$-and-$h$ in orange vs Gaussian in green) conditional probability density functions at four values of the scalar feature $x$, in the case where the target variable follows a Tukey $g$-and-$h$ distribution.

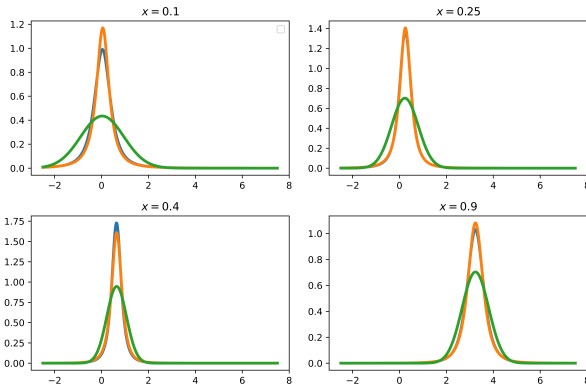

Figure 4: Comparison of true (blue) and fitted (Tukey $g$-and-$h$ in orange vs Gaussian in green) probability density functions at four values of the scalar feature $x$, in the case where the target variable follows a $t$-distribution.

The key areas of research in AI for agriculture include crop mapping, crop type mapping, field boundary delineation, yield estimation, and pests and disease detection. While there are other use cases such as crop

suitability, the aforementioned categories encompass the majority of research conducted in this field. We can determine yield at various scales, assess crop conditions, and combine the model outputs with other data sources, such as temperature and rainfall predictions, to derive comprehensive insights. These insights can aid in decision-making for farmers and policymakers, facilitating informed choices regarding yields, diseases, and other relevant factors.

We apply our proposed methodology to global crop yield prediction since food security in general and yield prediction in particular pose one of the most pressing issues in AI for agriculture. Extreme events can lead to crop yield declines, resulting in financial losses and threats to food security. Crop yield prediction involves predicting the volume or weight of crops to be harvested in each unit area. This is a regression problem where the goal is to estimate crop yield per unit area, representing the rate of production. There is extensive literature on crop yield prediction, where it can be considered on a plant, farm, global or regional scales. In this work, we consider a global scale and therefore utilise a global dataset for crop yield prediction. Shuai et al. predicted maize yield in the United States at the pixel level, providing estimates in tonnes per hectare (Shuai & Basso, 2022). The paper by You et al. (2017) introduced a different approach, employing deep Gaussian processes to predict crop yield based on remote sensing data (You et al., 2017), with focus on estimating yield at the county scale rather than at the individual pixel level.

We investigate a yearly global yield dataset (Iizumi, 2019) for maize, wheat, rice, and soybean, spanning the period 1981-2016. For each year and each crop, the yield is provided in tons per hectare on a global grid with 0.5' spatial resolution. For instance, Figure 5 shows the global yields for maize in 2010. Our goal is to learn, for each crop, a parametric distribution of the yield as a function of latitude, longitude, and year. We compare two approaches:

1. Gaussian prediction, based on a negative Gaussian likelihood loss function.

2. Tukey $g$-and-$h$ prediction, based on a negative Tukey $g$-and-$h$ likelihood loss function, as described in Section 2.2.

Our rationale for leveraging neural networks for this problem is that we expect a strongly non-linear dependence of the Tukey $g$-and-$h$ parameters with respect to latitude and longitude. Our neural networks take three inputs, the longitude, the latitude, and the year, and output the parameters of a probability distribution (specifically, 2 outputs for a Gaussian distribution, 4 outputs for the Tuley $g$-and-$h$ distribution). For each parametric distribution, we use the Optuna package to select hyperparameters (details of hyperparameters are provided in Appendix C), using 100 trials. Training is performed according to the Adam algorithm (Kingma & Ba, 2015), and with the use of a scheduler. We split the data into years 1985, 1995, 2005, 2015 for validation and 1986, 1996, 2006 for testing. All remaining years from the dataset are used for training.

In Table 1 we report the logarithmic score for each distribution over the validation and test datasets for maize. Results for other crops are available on the GitHub repository.

In Figure 6a we present a PIT histogram for the test data when training a Tukey $g$-and-$h$ distribution. Overall, we observe a good fit between the data and the model, except for the lower 5% values. In comparison, in Figure 7a we present a PIT histogram for the same test data when training a Gaussian distribution. There is a clear mismatch between the data and the model as we observe a clear departure from the uniform distribution.

In future work, we will investigate the use of weather indicators to more accurately predict the yield. Average and extreme weather conditions over the year naturally play a key role in determining the yield. Additionally, the spatial dependence of weather conditions results in spatial dependence of the yield, which we wish to remove as much as possible from a methodological point of view.

## 5 Conclusion

The Tukey $g$-and-$h$ distribution has a strong history of research and applications in environmental sciences and other fields such as financial modeling. It offers a good trade-off between a limited number of parame-

| method | year | LS | q01 | q05 | q25 | q50 | q75 | q95 | q99 |
|---|---|---|---|---|---|---|---|---|---|
| gaussian | - | -1.03 | 0.00 | 0.03 | 0.22 | 0.49 | 0.72 | 0.91 | 0.97 |
| tukey | - | -1.14 | 0.00 | 0.03 | 0.23 | 0.49 | 0.74 | 0.95 | 0.99 |
| gaussian | 1986 | -1.18 | 0.00 | 0.02 | 0.21 | 0.48 | 0.72 | 0.91 | 0.97 |
| tukey | 1986 | -1.30 | 0.00 | 0.03 | 0.22 | 0.48 | 0.73 | 0.95 | 0.99 |
| gaussian | 1996 | -0.99 | 0.00 | 0.02 | 0.20 | 0.46 | 0.69 | 0.89 | 0.96 |
| tukey | 1996 | -1.13 | 0.00 | 0.03 | 0.21 | 0.46 | 0.71 | 0.93 | 0.99 |
| gaussian | 2006 | -0.92 | 0.00 | 0.03 | 0.25 | 0.53 | 0.76 | 0.93 | 0.97 |
| tukey | 2006 | -1.00 | 0.01 | 0.04 | 0.25 | 0.52 | 0.77 | 0.96 | 0.99 |

Table 1: Quantitative comparison of training a Gaussian versus a Tukey $g$-and-$h$ distribution on the maize dataset, by evaluation on the test dataset. The rows with year indicated as - correspond to quantities evaluated across all test years. The quantities q[$100\alpha$] are defined by (16).

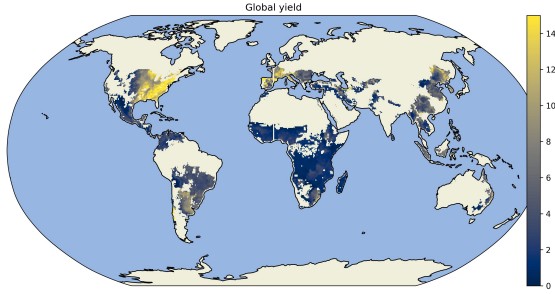

Figure 5: Global yield of maize in ton per hectare in 2010 on a 0.5' spatial-resolution grid (Iizumi, 2019).

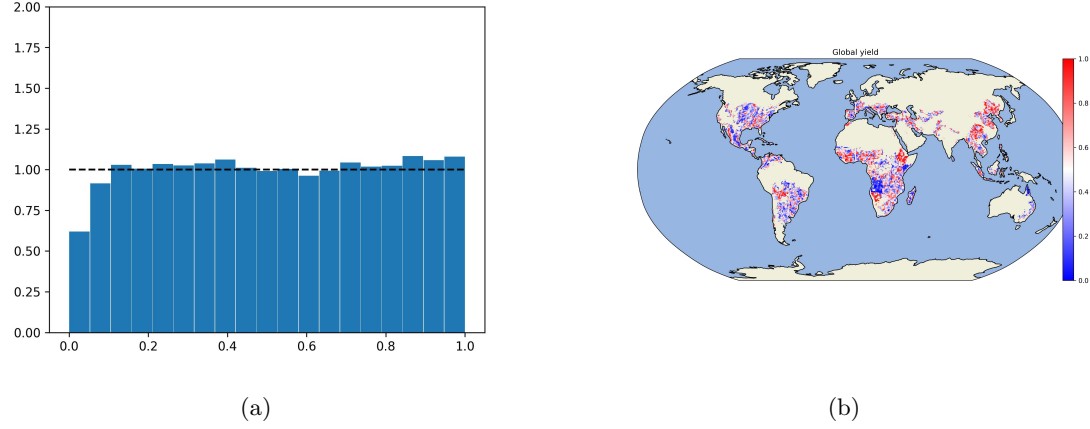

Figure 6: PIT residuals from maize crop yield test data for the Tukey $g$-and-$h$ distribution: (a) histogram (all test years); (b) global map (year 2006).

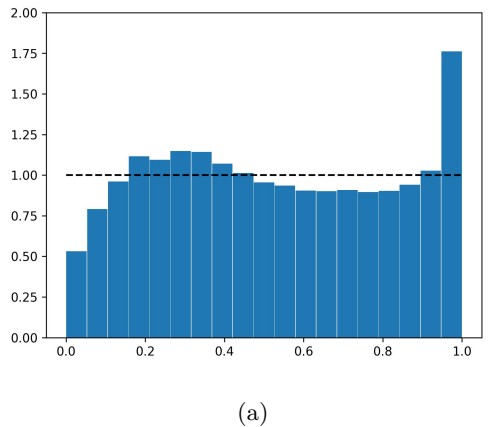 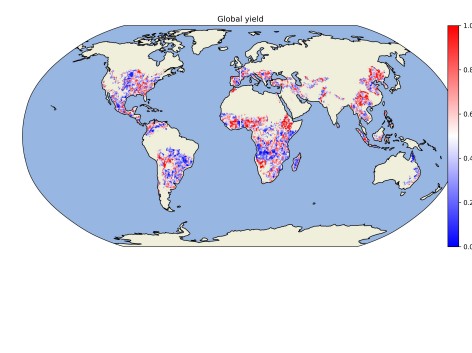

(a)                                   (b)

Figure 7: PIT residuals from maize crop yield test data for the Gaussian distribution: (a) histogram (all test years); (b) global map (year 2006).

ters and the ability to approximate a wide range of skewed and heavy-tailed probability density functions. Applications of Deep neural networks is also becoming more and more prevalent in the aforementioned fields. While the exact form of the Tukey $g$-and-$h$ log-likelihood has no known closed-form solution, we show in this paper that we can still train a Deep neural network to predict a Tukey $g$-and-$h$ distribution. While the standard approach in Tukey $g$-and-$h$ random fields is to assume that a unique Tukey $g$-and-$h$ is applied pointwise, here we allow the parameters of the transform to be features-dependent, and learn the corresponding mapping via standard neural network training techniques. This can naturally be extended to multi-modal Tukey $g$-and-$h$ distributions with 4 output neurons for each mode and a softmax over $p$ neurons where $p$ is the number of modes. A natural direction for future work would be to address the remaining dependence in the target variable. For instance, looking at Figure 6b, it is clear that there remains some spatial dependence between residuals at neighbouring locations. Additional features — such as weather patterns for this application— may further account for this remaining dependence. Another approach might be to account for these patterns of dependence via Gaussian Process regression.

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

## A   Simulation study — hyperparameter selection

In our simulation study, both for the Gaussian and the Tukey $g$-and-$h$ distributions, we carry out hyperparameter selection using the Optuna package. We use training and validation datasets to select hyperparameters by carrying out 50 trials. Table 2 provides the list of hyperparameters, their respective ranges, and whether they are sampled on a logarithmic scale.

| Hyperparameter | Range | Log-Scale |
|---|---:|:---:|
| Batch size | $2^8$ to $2^{12}$ | ✓ |
| Number of epochs | 50 (fixed) | — |
| Layer size | $2^4$ to $2^{10}$ | ✓ |
| Number of layers | 2 to 6 | — |
| Batch normalization | True, False | — |
| Learning rate | $10^{-5}$ to $10^{-1}$ | ✓ |
| Scheduler step size | 5 to 20 | — |

Table 2: summary of hyperparameters and their ranges

## B   Simulation study — Logarithmic Scores

In the tables below we present distribution summaries of the LS over the validation and test datasets for the Gaussian and Tukey $g$-and-$h$ methods. These summaries are computed over 10 independent simulated datasets. For each realization, training and validation data were leveraged to carry out hyperparameter selection using the Optuna package over 50 trials. The figures we report are obtained from the best hyperparameter combination for each of the independent realizations.

| method | validation | | | | test | | | |
|---|---|---|---|---|---|---|---|---|
| | mean | std | min | max | mean | std | min | max |
| gaussian | 0.523546 | 0.016915 | 0.500580 | 0.548912 | 0.636190 | 0.114233 | 0.473385 | 0.784348 |
| tukey | 0.419505 | 0.009559 | 0.405141 | 0.428925 | 0.450936 | 0.028840 | 0.417795 | 0.491850 |

Table 3: LS distribution summary statistics over 10 realizations for sample size 4000

| method | validation | | | | test | | | |
|---|---|---|---|---|---|---|---|---|
| | mean | std | min | max | mean | std | min | max |
| gaussian | 0.525525 | 0.015846 | 0.502940 | 0.548875 | 0.538332 | 0.082705 | 0.438079 | 0.726418 |
| tukey | 0.405474 | 0.006244 | 0.394731 | 0.415522 | 0.404037 | 0.023347 | 0.369288 | 0.436069 |

Table 4: LS distribution summary statistics over 10 realizations for sample size 40000

## C  Crop yield application — hyperparameter selection

For our application to global crop yield data, we again use the Optuna framework to select hyperparameters. Table 5 provides the list of hyperparameters and their ranges.

| Hyperparameter | Range | Log-Scale |
|---|---:|:---:|
| Batch size | $2^8$ to $2^{12}$ | ✓ |
| Number of epochs | 50 (fixed) | — |
| Layer size | $2^4$ to $2^{10}$ | ✓ |
| Number of layers | 4 to 8 | — |
| Batch normalization | True, False | — |
| Learning rate | $10^{-5}$ to $10^{-1}$ | ✓ |
| Scheduler step size | 5 to 20 | — |

Table 5: summary of hyperparameters and their ranges

## D  Computational efficiency

In Table 6 we present an analysis of the computational efficiency of stochastic gradient descent on the negative log-likelihood. As the size of the neural network increases, the additional computational cost incurred by the binary search inversion of the Tukey $g$-and-$h$ transform becomes negligible in comparison to the overall cost of backward propagation.

| Size | 2 Layers | | 3 Layers | | 4 Layers | | 5 Layers | |
|---|---|---|---|---|---|---|---|---|
| | Gaussian | Tukey | Gaussian | Tukey | Gaussian | Tukey | Gaussian | Tukey |
| 128 | 9.18e-04 | 1.05e-02 | 1.01e-03 | 1.06e-02 | 1.14e-03 | 1.07e-02 | 1.28e-03 | 1.06e-02 |
| 256 | 6.05e-04 | 5.22e-03 | 7.10e-04 | 4.32e-03 | 9.20e-04 | 4.34e-03 | 1.11e-03 | 4.80e-03 |
| 512 | 1.05e-03 | 4.57e-03 | 1.78e-03 | 5.32e-03 | 2.59e-03 | 7.40e-03 | 3.36e-03 | 8.03e-03 |
| 1024 | 2.81e-03 | 7.48e-03 | 5.32e-03 | 8.96e-03 | 7.74e-03 | 1.13e-02 | 1.01e-02 | 1.37e-02 |

Table 6: Mean computational time (Intel i7 CPU) of one step of Adam gradient descent on a batch size of 256 for a fully connected forward neural network with varying number of layers and layer size.

