# OpenReview forum: "Tukey g-and-h neural network regression for non-Gaussian data"
_TMLR — Rejected by TMLR_

### Review · Reviewer_gTbt · 2025-08-27

**Summary Of Contributions:**

This paper applies the "Tukey g-and-h transformation" to a standard normal random variable, resulting in a new random variable with a "G-and-H distribution." This transformation is an increasing function with four parameters that control the "shape" of the probability density. The author(s) leverage a neural network to predict these distribution parameters for each input. The proposed model is then applied to several regression tasks and trained by minimizing the negative log-likelihood.

+ Strengths
1. The paper is clearly written and quite easy to understand.
2. The experimental results on synthetic and real-world datasets show improved performance through the neural-network-controlled distribution.

+ Weaknesses
1. The novelty of the proposed method is limited. It essentially involves parameterizing a simple distribution (e.g., a Gaussian distribution) through neural networks, which has become a common practice in models like variational autoencoders and flow-based models.
2. The advantages of the proposed method are not sufficiently emphasized. Specifically, the author(s) only use the parametric Gaussian distribution as a baseline in the crop yield prediction experiment. The choice of the Tukey g-and-h transformation over other alternatives (e.g., another increasing function with learnable parameters) is not fully justified. I understand that the author(s) may want to demonstrate the greater flexibility offered by the parametric distribution. However, it would be helpful to include evidence of running time to show that this added flexibility does not significantly compromise computational efficiency.

**Audience:**

Yes

**Audience Explanation:**

This work will be of interest to TMLR readers who are concerned with probabilistic regression for non-Gaussian targets and practical uncertainty quantification. The global crop-yield application demonstrates relevance for climate/agriculture modeling. The authors note that a PyTorch implementation is available, which aids reproducibility and reuse.

**Broader Impact Concerns:**

I don't think there are any major ethical red flags in this work.

**Claims And Evidence:**

No

**Claims Explanation:**

The core method is sound. The paper clearly derives the negative log-likelihood and the implicit-function gradients using binary search, which supports the trainability and correctness of the loss. However, the empirical evidence is not yet sufficient, as the baselines are currently limited to a Gaussian likelihood. Since the title indicates that this work focuses on non-Gaussian data, including other non-Gaussian baselines (e.g., Student-t, normalizing flow) would help assess the claimed benefits.

The paper states that the inverse of the Tukey transformation "at any given point can be obtained efficiently by binary search," but it does not provide runtime statistics or complexity analysis to support this claim.

**Requested Changes:**

1. Including additional non-Gaussian or neural network-based baselines in the experiment section would be beneficial. Reporting results with negative log-likelihood and root mean squared error, along with standard errors across multiple random seeds, will better highlight the advantages of the proposed method.

2. The potential computational efficiency can be demonstrated in a table comparing running times.

3. Conducting more ablation and sensitivity studies, such as varying the network architecture, can better justify the design choices and help readers understand the range of applications of this method.

---

> ### Author Response · Authors · 2025-11-01
> **Response to review**
>
> Thank you for your review and in particular for your suggestion to carry out "more ablation and sensitivity studies, such as varying the network architecture". We have decided to achieve this using the Optuna package. We aimed at providing details of the hyperparameters over which we optimize on the validation loss directly in the paper. While all the code will be available on the authors' github page (not currently shared in the manuscript due to double-blind), if the details provided in the paper are not clear enough please let us know. In addition, the computational study is now carried out over 10 independent datasets.
>
> As for the computational study, we agree that it was lacking from the previous version: as we mentioned in another comment, it is natural to ask whether the computational cost due to the binary search inversion hinders the overall feasibility of the method for practical applications. Our computational study aims at demonstrating that the cost of the inversion of the Tukey g-and-h transform rapidly becomes negligible in comparison to the overall cost of backpropagation as the neural network size increases.
>
> As per your other request, we also now report the logarithmic score (negative log likelihood). We did not report the Mean Square Error as we do not necessarily expect much benefit from using the Tukey g-and-h if the key interest is to indeed minimize the Mean Square Error of a point prediction. Here our interest is more drawn towards applications of probabilistic prediction.
>
> In terms of additional non-Gaussian methods, we agree that there are other methods available. We expect however that normalizing flows would be more appropriate for high-dimensional target data. For scalar target data as in the application considered here, unless the data were for instance multi-modal, we expect that a parametric distribution such as the Tukey g-and-h would be a more natural fit. Another alternative candidate method would be quantile regression, as mentioned in the paper. The relevant method will depend on the application: if one is interested in a specific quantile, quantile regression might be a better approach, except for extreme quantiles (by extreme we mean towards the tails of the distribution, where by definition there is less data and therefore non-parametric methods tend to give poor results).
> As for choosing another transformation than the Tukey g-and-h, this is a valid remark, but probably beyond the scope of the paper: the Tukey g-and-h transformation has become popular in fields such as geosciences and finance, and we aim to extend its application to neural network regression.
>
> Please let us know whether this addresses your queries.

---

### Review · Reviewer_227d · 2025-09-17

**Summary Of Contributions:**

The authors study a non-parametric regression model that combines a Tukey g-and-h distribution with a series of deep neural networks. Indeed, the four parameters in the Tukey g-and-h distribution are replaced by neural networks. The input of the neural networks is some spatial domain turning a sample of the model into a spatially dependent random object. The authors explain how the corresponding networks should be trained in practice given regression data. They provide two simulated toy examples and an application in crop yield predictions.

Strengths:
- the Tuckey g-and-h distribution appear to be a much more flexible framework compared to usual Gaussian distributions
- real data experiment

Weakness:
- the experiment section is very short, lacks detail and not persuasive
- the methodology is not studied very deeply
- not all of the computational ideas are very persuasive, such as the use of binary search to invert $\tau$, instead of Newton's method

**Audience:**

Yes

**Audience Explanation:**

Some people could be interested in using the Tukey g-and-h distribution in machine learning. However, this paper may not provide enough information and new insights to find much interest.

**Broader Impact Concerns:**

N.A.

**Claims And Evidence:**

No

**Claims Explanation:**

The authors write in the abstract that they 'demonstrate the efficiency of [their] procedure in simulated examples'. However, they actually only provide a single example in each of the simulated examples using one data set each. To get a better feeling on how effective and efficient the method is, I would expect to test the method on a multitude of underlying test problems -- i.e. different functions that generate the data, resampled data sets, and averaged outcomes. The authors should also compute error metrics instead of only plotting graphs.

Moreover, they do not provide enough information for the reader to validate the experiments. They speak of 'arbitrary' functions rather than specifying their precise test case.

**Requested Changes:**

- the authors should consider Newton's method instead of binary search for the inversion of $\tau$, as it will converge considerably faster -- they could add a comparison study
 - the authors should add a more significant study of simulated experiments, not only including different data generating distributions, resampled datasets, but also different numbers of data sets to show the robustness of their method
 - in addition to Gaussian neural network regression, the authors should compare their method to Gaussian process regression
 - the authors should make sure that their paper/appendix contains sufficient information to validate their results

---

> ### Author Response · Authors · 2025-10-31
> **Response to review**
>
> Thank you for your review. And for suggesting the use of Newton's method. We are currently investigating its implementation for this problem, but have run in what seems to be convergence issues. This is potentially due to the derivative becoming very small for values of g and h far from the true values. While we are keen to get this to work as it would provide additional computational gains, we have currently added a short computational study that shows that the cost of the inversion, when done via binary search, becomes negligible in comparison to to the overall cost of backpropagation. As per your suggestion, Newton's method would further reduce this additional cost for small size neural networks.
>
> Following your requested change of a more significant study, we have made significant changes - please see the newly submitted revision and its comments.
>
> You also made a comment regarding the lack of quantitative metrics provided in the paper and its appendix. We agree with that observation, and hopefully the submitted revision should address this. We now leverage the logarithmic score as the metric we report, as per some litterature on probabilistic prediction, which we refer to in the new manuscript.
>
> Regarding reproducibility, the code and datasets will be made available on the author's github (not currently shared so as not to break double-blind). Yet we aimed to provide further explanation on how the simulation study is carried out, in particular regarding the selection of hyperparameters.
>
> Please let us know if there are any further developments you would like to see in the simulation study.

---

### Review · Reviewer_VSBG · 2025-09-26

**Summary Of Contributions:**

- The paper introduces a regression method using neural networks with the Tukey g-and-h distribution to handle non-Gaussian data.
- The Tukey g-and-h transform flexibly introduces skewness and kurtosis, approximating distributions like Cauchy and Student-t.
- The model trains by predicting distribution parameters and minimizing the negative log-likelihood, which lacks a closed form.
- The method is validated on simulations and applied to real-world global crop yield data across multiple crops.
- A goodness-of-fit analysis is performed, and the authors provide both a PyTorch implementation on GitHub and a PyPI package.

**Audience:**

Yes

**Audience Explanation:**

- The work addresses interesting and impactful real-world problems, such as modeling global crop yields.
- The authors provide practical resources, including a PyTorch implementation and PyPI package, which enhance reproducibility and usability for the research community.

**Broader Impact Concerns:**

There are no broader impact concerns on this work.

**Claims And Evidence:**

No

**Claims Explanation:**

- It demonstrates a well-motivated and effective application of the Tukey g-and-h transform in a neural network regression setting.
- The paper does not sufficiently justify why the Tukey g-and-h transform is the most suitable choice for the considered problems compared to alternative flexible distributions.
- The application of the Tukey g-and-h transform feels somewhat straightforward, and the paper does not adequately highlight the technical challenges involved or how they were overcome.
- The statement that “the main challenge lies in the inversion of the Tukey g-and-h transform” is not clearly explained, and the reader is left unsure why this inversion is problematic in evaluating the negative log-likelihood.
- The goodness-of-fit analysis section lacks clarity, and it is not explained why common measures such as KL divergence are not applicable or were not chosen.

**Requested Changes:**

- Please consider revising the points raised above.
- The overall writing could be improved for clarity and accessibility, as some sections are hard to follow or leave important details underdeveloped; please see below.

---

- The notation g-and-h should consistently appear as $g$-and-$h$.
- Similarly, student-t should be written as student-$t$.
- The names of libraries should be properly capitalized: PyTorch, GitHub, and PyPI.
- On Page 3, G-and-H should be corrected to g-and-h.
- Some capitalizations are unnecessary and distracting. For example, Stochastic Gradient Descent and Machine Learning should not be capitalized.
- Section 4 feels overly redundant; while it is important to highlight the real-world problems tackled, the section could be made more concise.
- On Page 9, a follows should be corrected to as follows.

---

> ### Author Response · Authors · 2025-10-31
> **Response to review**
>
> Thank you for your review. We have submitted a new version that should hopefully address a few points you have mentioned, including your remarks regarding notation. In particular, we have rewritten the section on goodness-of-fit.
>
> "The paper does not sufficiently justify why the Tukey g-and-h transform is the most suitable choice for the considered problems compared to alternative flexible distributions."
>
> We do not claim that the Tukey g-and-h is necessarily the best available parametric distribution among all existing distributions for this problem. There might be another distribution whose properties are better-suited to this problem. The benefit of the Tukey g-and-h distribution, beyond the scope of this paper, is that it provides a good approximation to other standard probability distributions, while offering a simple parametric form. This has made this distribution quite popular in geosciences and finance. The aim of this paper is to extend the scope of applications for the Tukey g-and-h transform to neural network regression.
>
>  "The application of the Tukey g-and-h transform feels somewhat straightforward, and the paper does not adequately highlight the technical challenges involved or how they were overcome.
> The statement that “the main challenge lies in the inversion of the Tukey g-and-h transform” is not clearly explained, and the reader is left unsure why this inversion is problematic in evaluating the negative log-likelihood."
>
> We agree that there is nothing fundamentally complicated in this paper. For us, this is a good reason for the Tukey g-and-h distribution to be applied more widely. Regarding the inversion, it is not complicated per say. However, one might think that the binary search would be computationally too costly (we have had such questions in previous presentations of this work). However, and we show this in a short computational study in the appendix of the new version, the inversion cost quickly becomes negligible in comparison to the overall cost of the backpropagation, as the size of the neural network grows. Another reviwer also suggested replacing the binary search by Newton's method for the inversion: we are currently investigating this, as it would indeed provide further computational gains for smaller neural networks, but have currently run in convergence issues.
>
> Regarding your saying "the paper does not adequately highlight the technical challenges involved or how they were overcome", is there any particular point you had in mind, beyond the section on goodness-of-fit which you mentioned separately?
>
> Please let us know if you would like us to further clarify some of the above points.

---

### Decision · Action_Editor_69DN · 2026-01-05

**Recommendation:** Reject

**Audience:**

No

**Audience Explanation:**

Without a proper experimental evaluation, as mentioned above, it is challenging to see how readers could be interested in building on top of this method or using it in practice.

**Claims And Evidence:**

No

**Claims Explanation:**

I appreciate the authors' efforts to provide new experiments after the initial reviews. However, I completely agree with reviewer gTbt. The paper needs to include baselines beyond the Gaussian baseline currently present. At least, the inclusion of a Bayesian neural network (BNN) would be the minimum baseline. There are several software implementations of BNNs out there that could be used off the shelf. Otherwise, it is hard to pinpoint the contributions of the method. In the abstract, the authors mention that the Tukey g-and-h distribution has found applications in finance and geoscience. Additional experiments with real datasets showing what the method addresses in these applications, compared to methods used in those communities, are needed to understand the relative usefulness of this new methodological development. There is only an experiment with real data.